# NAM - Unsupervised Cross-Domain Image Mapping without Cycles or GANs

**Yedid Hoshen, Lior Wolf**
{yedidh, wolf}@fb.com

## Abstract

Several methods were recently proposed for Unsupervised Domain Mapping, which is the task of translating images between domains without prior knowledge of correspondences. Current approaches suffer from an instability in training due to relying on GANs which are powerful but highly sensitive to hyper-parameters and suffer from mode collapse. In addition, most methods rely heavily on "cycle" relationships between the domains, which enforce a one-to-one mapping. In this work, we introduce an alternative method: NAM. NAM relies on a pre-trained generative model of the source domain, and aligns each target image with an image sampled from the source distribution while jointly optimizing the domain mapping function. Experiments are presented validating the effectiveness of our method.

## 1 Introduction

Humans can easily imagine how a scene observed by day, would appear at night. This is an instance of mapping across domains. This ability is not limited to domains such as day and night, for which exact correspondence can be obtained simply by revisiting the same scene after a few hours. Humans can also visually adapt domains between which they have never seen any correspondences such as, imagining how a Picasso painting would appear if painted by Rembrandt. This motivates Unsupervised Domain Mapping, visually mapping between domains when no correspondences are given between samples in the training set. Unsupervised Domain Mapping typically operates by finding a function for mapping images between the domains so that after mapping, the distribution of mapped source images is identical to that of the target images.

Due to its scientific and practical importance, Unsupervised Domain Adaptation has recently attracted significant interest from the research community, particularly for visual domains. Successful recent approaches, e.g. DTN (Taigman et al., 2017), CycleGANs (Zhu et al., 2017) and Disco-GAN (Kim et al., 2017), utilize Generative Adversarial Networks (GANs) to model the distributions of the two domains, $\mathcal{X}$ and $\mathcal{Y}$. GANs are very effective tools for generative models of images, however they suffer from instability in training, making their use challenging. The instability typically requires careful choice of hyper-parameters and often multiple initializations due to mode collapse. Current methods also make additional assumptions that can be restrictive, e.g., DTN assumes that a pre-trained high-quality domain specific feature extractor exists which is effective for both domains. This assumption is good for the domain of faces (which is the main application of DTN) but may not be valid for all cases. CycleGAN and DiscoGAN make the assumption that a transformation $T_{XY}$ can be found for every $\mathcal{X}$-domain image $x$ to a unique $\mathcal{Y}$-domain image $y$, and another transformation $T_{YX}$ exists between the B domain and the original A-domain image $y = T_{XY}(x), x = T_{YX}(y)$. This is problematic if the actual mapping is many-to-one or one-to-many, as in super-resolution or subsampling.

We suggest a novel method for removing the reliance on GANs for domain mapping. Our algorithm leverages a generative model of domain $\mathcal{X}$ to synthesize a matching image $x$ for every image $y$ in domain $\mathcal{Y}$ such that $y = T(x)$, where $T()$ is a learned mapping function.

Our method consists of 3 parts:

1. An accurate parametric model for the domain $\mathcal{X}$. This model, $G(z)$, is parametrized by a vector $z$. The model is trained using some state-of-the-art method including: GLO (Bo-

janowski et al., 2017), VAE (Kingma & Welling, 2013), GAN (Goodfellow et al., 2014) or a differentiable hand-crafted physical simulation.

2. A mapping function $T()$ which translates images from the $\mathcal{X}$ domain to the $\mathcal{Y}$ domain. The mapping is learned as a part of our algorithm.

3. A set of target images $y \in \mathcal{Y}$. For each target image $y$ we find an image $G(z)$ such that $y = T(G(z))$.

Our model is very different from existing unsupervised image-to-image mapping models in that no adversarial training takes place when mapping between the domains and so doing escapes the need to use cycles or GANs (although GANs can optionally be used for obtaining a generative model for the $\mathcal{X}$ domain).

## 2 Unsupervised Image Mapping without GANs

In this section, we present our method - NAM - for unsupervised domain mapping. Our method is related to GLO by Bojanowski et al. (2017) which we shall briefly highlight.

### 2.1 Generative Latent Optimization (GLO)

GLO was recently introduced by Bojanowski et al. (2017). Let us define the set of training images as $\{x \in \mathcal{X}\}$. For each image $x$, we learn a latent representation $z_x$ and we jointly learn a general generator function $G(.)$, which takes as input a latent representation $z'$, and generates an image $G(z')$. The objective is to optimize $G(.)$ and the set of latent representations $\{z_x \in Z\}$ such that they recover the training images.

The optimization is performed as follows:

$$argmin_{G,z_x} \sum_x L_p(G(z_x), x), \tag{1}$$

where $L_p$ is the Laplacian pyramid loss, which was found to be significantly better than the Euclidean loss for this purpose.

### 2.2 Non-Adversarial Mapping (NAM)

Differently from GLO we tackle the task of unsupervised image mapping between domains. We are given sets of images $X$ and $Y$, without correspondences. Our task is to obtain for every image $y \in Y$, an image $x'$ that appears to come from domain $\mathcal{X}$ and that the mapping $T(x')$ is similar to $y$.

We assume that a generative model of domain $\mathcal{X}$ was obtained by a previous method. This method can be GLO (Bojanowski et al., 2017), VAE (Kingma & Welling, 2013), GAN (Goodfellow et al., 2014) or a hand designed simulator (see for example Wolf et al. (2017)). The generative model of $\mathcal{X}$ consists of a function $G(.)$ which for every $z$, yields image $G(z)$ which appears to come from domain $\mathcal{X}$.

We introduce mapping function $T(.)$ which will be trained during the optimization of NAM. $T(.)$ takes an $\mathcal{X}$ domain image and maps it to an image appearing to come from the $\mathcal{Y}$ domain.

For every image $y \in Y$, we find latent vector $s_y$ such that $T(G(s_y)) = y$. This is achieved by both training domain mapping function $T(.)$ as well as latent vector $s_y$ for every $y \in Y$. The $X$ generative model $G()$ is kept fixed.

$$argmin_{T,s_y} \sum_y L_{VGG}(T(G(s_y)), y), \tag{2}$$

Differently from GLO, we use a VGG perceptual loss rather than the Laplacian pyramid. The VGG perceptual loss was found by several recent papers (Chen & Koltun, 2017; Zhang et al., 2018) to give perceptually pleasing results.

Table 1: NAM (center row) vs. DiscoGAN/CycleGAN (top row) for a given input (bottom row).

| Edges2Handbags | Edges2Shoes |
|---|---|

| SVHN→MNIST | MNIST→SVHN |
|---|---|

NAM mapping from a single source image (shown last) for different random initializations

## 3 EXPERIMENTS

We evaluate our method NAM against DiscoGAN/CycleGAN on several datasets: Edges2Handbags, Edges2Shoes, SVHN → MNIST and MNIST → SVHN. For each column of images: the image mapped by the baseline (DiscoGAN for Edges2Shoes/Handbags and CycleGAN for SVHN and MNIST) is shown on the top row. The image mapped by NAM is shown in the center, the source image is shown at the bottom. From looking at the results we can see that NAM is better at rendering textures. Additionally CycleGAN failed on (MNIST → SVHN) whereas NAM had more success, as it does not rely on cycle constraints. Generally, the images mapped by NAM appear more realistic and have fewer holes, probably due to the superior quality of the pre-trained generative model rather the one trained on the fly in the baseline domain mapping. Another attractive property of NAM is the variability in mapping for the same image by multiple random initializations.

## 4 DISCUSSION

Unsupervised mapping between domains is an exciting technology with many applications. While existing work is currently dominated by adversarial training, we present results that support other forms of training.

Adversarial constraints operate at the distribution level, while circularity based work, such as Zhu et al. (2017); Kim et al. (2017), augment these constraints with bidirectional per-sample constraints. Our work focuses entirely on unidirectional per-sample constraints.

Our method relies on having a high quality pre-trained unsupervised generative model for the $\mathcal{X}$ domain. With the recent advent of very high resolution generative models, e.g., by Karras et al. (2017), our method can be scaled to very large images.

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

# A  IMPLEMENTATION DETAILS

In this section we give a detailed description of the procedure used to create the results shown in the experimental section of this abstract.

$\mathcal{X}$ *domain generative model* $G(.)$*:* Our method takes as input a pre-trained generative model for the $\mathcal{X}$ domain. Although many choices of generative models are available, we opted for the most commonly used. Therefore all $G(.)$ used in our experiments were trained with the DCGAN model. We used the PyTorch code released by Radford et al. (2015). For Edges2Shoes (Yu & Grauman, 2014) and Edges2Handbags (Zhu et al., 2016) we used 100 latent dimensions, for SVHN (Netzer et al., 2011) and MNIST (LeCun & Cortes, 2010) we used 32 latent dimensions. All other hyperparameters were unchanged from the GitHub code. For some datasets the DCGAN experienced divergence after about 30 epochs, in such cases the model checkpoint before the divergence occurred were selected.

For completeness we mention that we also performed the experiments with GLO (Bojanowski et al., 2017) based $G(.)$. The method worked successfully however the visual quality of the resulting $G()$ did not surpass that of DCGAN.

*Mapping function* $T(.)$*:* There were several considerations in the design of the mapping function:

1. Being powerful enough to describe the mappings between the domains.

2. Not being too large as to overfit.

3. Being able to model the similar behavior in different areas of the spatial pyramids in corresponding images in the two domains.

We elected to use a network with an architecture based on Chen & Koltun (2017). We found that quite small networks achieved better matches between the domains (as visually evaluated between $G(s_y)$ and $y$).

For the network accepting $64X64$ inputs (used on the Edges2Shoes and Edges2Handbags datasets), we used an architecture starting from a $4X4$ downsampled input image. The first layer consisted of 128 channels, all convolutions had support of $3X3$. The number of channels was halved for every layer in the network. All convolutional layers were followed by BatchNorm (Ioffe & Szegedy, 2015), and a Leaky ReLU non-linearity with negative slope of $0.2$. The output of each layer was upsampled bilinearly by a factor of 2. It was then concatenated with the input image downsampled to the appropriate resolution. This process was repeated until the final resolution ($64X64$) was achieved. A final convolutional layer reduced the channel count to 3.

The $T(.)$ function used in the $32X32$ (MNIST and SVHN) experiments was identical to the above expect for using only 32 channels in the first layer, and scaling only up to $32X32$ (one fewer layer).

*Optimization:* We carried out SGD optimization using the ADAM (Kingma & Ba, 2016) method. For all datasets we used a learning rate of $0.03$ for the latent codes $z_x$ and $0.001$ for the mapping function $T(.)$. The difference in learning rate is due to $T(.)$ being updated every batch but any given $s_y$ is updated once an epoch. Momentum was set to $0.5$ and no weight-decay was used. 50 training epochs were used for the Edges2Shoes dataset and 110 epochs for all other datasets (results were not very sensitive to the number of training epochs as long as it was larger than 40).

On all datasets training was performed on 2000 randomly selected examples from the $\mathcal{Y}$ domain. Using larger training sets was not found to be helpful, due to the further increasing the disparity in update rate between the $T(.)$ and every $s_y$. It is possible that by changing the training procedure it would be possible to use larger training sets, but this procedure worked best in our experiments.

*Generating results:* To map a $\mathcal{Y}$ domain image $y$, we calculated $s_y$ and presented $G(s_y)$ as the result of the method. The results of $G(s_y)$ (for mapping $Y \rightarrow X$) were typically better than those obtained $T(x)$ (for mapping $X \rightarrow Y$), due to the weak architecture selected for $T(.)$. In case that a strong $T(.)$ is required, we suggest to calculate a set of matching $G(s_y)$ and $y$ obtained using the procedure described above, and training a network $T(.)$ with a large capacity architecture using a fully supervised technique (e.g. as described by Chen & Koltun (2017)). A similar procedure was carried out in Hoshen & Wolf (2018).

*Multiple mappings for the same $\mathcal{Y}$ domain image $y$:* To obtain multiple mapped images for the same input image, we first trained function $T(.)$ (as well as $s_y$ for the training set) as described in the paper. For a new image $y$, we solved for $s_y$ that minimizes the loss $\|T(G(s_y)), y\|_{VGG}$ multiple times, each with $s_y$ initialized with a random normal distributed value. In all runs $T(.)$ and $G(.)$ were fixed and only $s_y$ was trained. Different runs displayed significant variation in $G(s_y)$, while (typically) appearing to be plausible mappings of $y$.

