# OpenReview forum: "NAM - Unsupervised Cross-Domain Image Mapping without Cycles or GANs"
_ICLR.cc/2018/Workshop — Accept_

### Official Review · AnonReviewer2 · 2018-03-09
**Accept**

**Rating:** 7
**Confidence:** 4

**Review:**

This workshop paper introdcues a new domain mapping approach which relies on generative models pretrained on the source domain, and aligns with  target images with an image sampled from the source domain. The joint optimization carried out during this mapping on the mapping function underlines the contribution.

The results are inline with the expectation from the method, and the contribution is on par with the workshop track. I suggest acceptance.

---

### Official Review · AnonReviewer3 · 2018-03-10
**nice work but lack quantitative results**

**Rating:** 4
**Confidence:** 4

**Review:**

This paper proposed a simple method for unsupervised cross domain image mapping. Different with the existing models, the model does not rely GANs, hence is not sensitive to mode collapse.  One drawback is that no quantitative results are shown in this paper.

---

### Official Review · AnonReviewer1 · 2018-03-13
**A simple idea that lacks experimental results support**

**Rating:** 6
**Confidence:** 2

**Review:**

 Unsupervised Domain Mapping is a very interesting topic with both theoretic and practical significance. Current approaches may suffer from an instability in training by using GANs (which are sensitive to hyper-parameters and model collapse). The authors of this work introduce so called non-adversarial mapping, which is based on a revision of the GAL framework proposed by Bojanowski et al. The main idea is to plug in a generative function learned from the source domain as a way to change the latent variables in the target domain, so that a mapping function T() is further used align them together.

The idea appears useful, however, it would be interesting if the authors could discuss in more detail how the mapping T() tends to align the distribution of the two domains. In case z (X domain hidden variable) and sy (Y domain hidden variable) have different concentrations (ie distribution), the mapping can be unstable compared with considering aligning the two distributions directly. Besides, the experimental results are very limited and only shown in terms of visual comparisons. which is difficult to fully demonstrate the usefulness of the proposed method.

---

### Decision · Program_Chairs · 2018-03-20
**ICLR 2018 Workshop Acceptance Decision**

**Decision:**

Accept

**Comment:**

Congratulations, your paper was accepted to the ICLR workshop.